# Unlocking Potential: A Comprehensive Overview of Cell Culture Banks and Their Impact on Biomedical Research

**DOI:** 10.3390/cells13221861

**Published:** 2024-11-10

**Authors:** Sabine Weiskirchen, Antonio M. Monteiro, Radovan Borojevic, Ralf Weiskirchen

**Affiliations:** 1Institute of Molecular Pathobiochemistry, Experimental Gene Therapy and Clinical Chemistry (IFMPEGKC), University Hospital Aachen, D-52074 Aachen, Germany; sweiskirchen@ukaachen.de; 2Banco de Células do Rio de Janeiro, Rio de Janeiro 25250-020, Brazil; amonteiro@hucff.ufrj.br (A.M.M.); rrborojevic@gmail.com (R.B.)

**Keywords:** cell culture banks, quality control, biomedical research, biobanking technology, ICLAC, cell authentication, contamination, immortalized cell lines, drug development, 3R principle

## Abstract

Cell culture banks play a crucial role in advancing biomedical research by providing standardized, reproducible biological materials essential for various applications, from drug development to regenerative medicine. This opinion article presents a comprehensive overview of cell culture banks, exploring their establishment, maintenance, and characterization processes. The significance of ethical considerations and regulatory frameworks governing the use of cell lines is discussed, emphasizing the importance of quality control and validation in ensuring the integrity of research outcomes. Additionally, the diverse types of cell culture banks—primary cells, immortalized cell lines, and stem cells—and their specific contributions to different fields such as cancer research, virology, and tissue engineering are examined. The impact of technological advancements on cell banking practices is also highlighted, including automation and biobanking software that enhance efficiency and data management. Furthermore, challenges faced by researchers in accessing high-quality cell lines are addressed, along with proposed strategies for improving collaboration between academic institutions and commercial entities. By unlocking the potential of cell culture banks through these discussions, this article aims to underline their indispensable role in driving innovation within biomedical research and fostering future discoveries that could lead to significant therapeutic breakthroughs.

## 1. Introduction

In the rapidly evolving field of biomedical research, the availability and reliability of biological materials are paramount. Cell culture banks have emerged as essential repositories that provide standardized and reproducible cell lines crucial for a myriad of applications, ranging from drug discovery to regenerative medicine [1]. These banks serve as foundational resources that enable researchers to conduct experiments with consistent quality and reliability, thereby facilitating advances in understanding diseases, developing therapies, and exploring novel scientific frontiers.

The concept of cell culture dates back to the beginning of the 20th century when scientists began isolating cells from living organisms for study [2]. Over time, advancements in technology and methodology have transformed this practice into a sophisticated discipline characterized by the establishment of cell culture banks. These banks not only preserve various types of cells but also ensure their viability and functionality over extended periods. This has significant implications for reproducibility in research findings, which is increasingly recognized as a cornerstone of scientific integrity. Moreover, newborn stem cell banking has become an important issue in personalized medicine and regenerative medicine [3]. Due to the necessity of cell lines in biomedical research, several widely known cell banks have been established in America, Europe, Asia, and Australia, which now serve as an effective source for high quality cells.

Cell culture banks offer several advantages that contribute to their importance in biomedical research. First and foremost, they provide access to well-characterized cell lines that have been thoroughly vetted for genetic stability, growth characteristics, and functional properties. This standardization minimizes variability in experimental outcomes caused by differences in cell origin or handling methods. Furthermore, these banks facilitate large-scale distribution of cell lines to researchers worldwide, promoting collaboration and accelerating the pace of discovery [1].

Another critical aspect is the ethical considerations surrounding the use of human-derived cells. As awareness grows regarding ethical sourcing and usage practices, adherence to guidelines established by regulatory bodies becomes increasingly vital. Cell culture banks often operate under stringent ethical standards that govern consent processes for obtaining human tissues or cells. By ensuring compliance with these regulations, they help maintain public trust in scientific research while also enabling studies that might otherwise be hindered by ethical dilemmas [4,5,6].

Nevertheless, despite their numerous advantages, cell culture banks can sometimes perpetuate issues related to genetic drift, contamination, and ethical concerns regarding the sourcing of cell lines. These issues can ultimately compromise the validity and reproducibility of research outcomes.

## 2. Expected Services of a Cell Bank Repository

A cell bank is expected to provide a reliable and standardized repository of biological materials, including various types of cell lines such as primary cells, immortalized cell lines, and stem cells. Additionally, a reputable cell bank should offer comprehensive documentation regarding the origin, handling protocols, and ethical sourcing of the cells (Figure 1).

Access to these resources promotes reproducibility in experiments and encourages collaboration within the scientific community, ultimately leading to advancements in biomedical research and therapeutic development. The following sections will discuss some aspects of the services expected from a cell bank repository.

### 2.1. Cell Line Catalog

A comprehensive cell line catalog serves as an essential resource for researchers by providing detailed information on each cell line’s characteristics. By including basic information, origin history with references, characterization data, authentication results, contamination testing outcomes, culture conditions, storage recommendations, functional characteristics, genetic characteristics, applications, ethical considerations, availability status, biosafety information, and support contact information, researchers can ensure high standards in research integrity and reproducibility. Such thorough documentation will facilitate more effective utilization of these invaluable tools in advancing scientific knowledge.

The integration of new IT methods, such as artificial intelligence and blockchain technology, is revolutionizing the management and accessibility of cell and tissue biobanks [7]. Blockchain technology is a decentralized and distributed digital ledger system that enables secure and transparent record-keeping of transactions across a network of computers. Each transaction is grouped into a block, which is then linked to the previous block, forming a chronological chain—hence the name “blockchain.” Key features of blockchain technology include decentralization, which means that, unlike traditional databases managed by a central authority, blockchain is maintained by a network of nodes, making it less susceptible to manipulation or failure. It offers a secure and transparent framework for tracking the provenance and usage of biological materials, ensuring compliance with ethical standards and enhancing trust among stakeholders, security, integrity, and traceability of samples in biobanks [8]. Together, these innovations not only improve operational efficiency but also promote greater accountability and transparency in biobanking practices. Moreover, transparency is a crucial aspect, as all transactions on a blockchain are visible to all participants in the network, fostering trust and accountability. Another important characteristic is immutability; once a block is added to the chain, it cannot be altered without consensus from the network, ensuring a permanent and tamper-proof record.

In addition to blockchain technology, various artificial intelligence (AI) techniques can significantly enhance the efficiency and effectiveness of cell culture biobanks. For instance, machine learning algorithms can analyze vast datasets to authenticate cell lines, identify optimal culture conditions, and predict cell viability, facilitating the development of customized protocols [9,10,11]. Additionally, natural language processing (NLP) can streamline the extraction of relevant information from scientific literature and historical records, aiding in data curation and improving decision-making processes within biobanks [12]. By employing AI, researchers can create an integrated catalog that streamlines data retrieval and enhances the searchability of biobank samples, facilitating quicker and more efficient research collaborations.

### 2.2. Authentication of Cell Lines

Authentication is essential to ensure that researchers work with reliable cell lines capable of producing valid results. By employing robust methods like short tandem repeat (STR) profiling, researchers can maintain high standards in scientific investigation and contribute valuable knowledge to their fields. If a cell bank upholds these practices, it not only enhances individual studies but also strengthens the integrity of biomedical research as a whole [13].

Currently, one of the most common techniques used by cell banks to detect cross-contamination and misidentification of cells is short tandem repeat (STR) profiling. This technique allows for comparing allele repeat counts at specific loci in DNA across different samples [14]. Although allelic variations in these repeats are typically polymorphic, the total number of alleles is limited. As a result, multiple STR loci are analyzed simultaneously using a multiplex PCR assay, which improves the effectiveness of STR profiles for identification or differentiation with strong statistical power. In STR analysis, the amplified variable microsatellite regions from the original DNA template are processed through a genetic analyzer. The data are then assessed with specialized software that determines the repeat numbers at each variant location (Figure 2).

Similarity search tools, such as CLASTR, can be used to compare resulting STR profiles with those available in specialized cell line knowledge resources such as the Cellosaurus database [15]. Today, numerous effective and standardized STR panels have been developed for various species and international recommendation for human and mouse cell lines have been established [13,15,16,17,18].

Some cell biobanks conduct chromosome analysis or perform spectral karyotyping (SKY) analysis for some special cell lines as a quality control (Figure 3). These analyses provide information about the number and structure of chromosomes, offering essential insights into genetic stability and abnormalities. SKY analysis enhances this process by using fluorescent probes to distinguish between different chromosomes, allowing for a more detailed examination of chromosomal arrangements and potential aneuploidies [19,20]. These services are crucial in estimating the genetic integrity of a cell line.

### 2.3. Contamination and Quality Control

Contamination in cell culture is a significant concern that can compromise experimental results and the integrity of research. One of the most common culprits is mycoplasma, a type of bacteria that lacks a cell wall and can easily infiltrate cell cultures without being detected through standard microscopy techniques [21]. Mycoplasma contamination can lead to altered cellular behavior, including changes in growth rates, metabolic functions, and gene expression profiles, ultimately skewing experimental outcomes [22]. Moreover, these contaminants are notoriously difficult to eradicate once established in a culture. To address this issue, reputable cell banks implement stringent testing protocols to screen for mycoplasma and other contaminants before distributing their cultures. The repertoire of methods used to detect contamination by mycoplasmas is large (Figure 4).

PCR-based methods focus on amplifying conserved segments of 16S rDNA or the space located between the 16S and 23S rDNA [21,23]. These assays use specialized oligonucleotide primers to simultaneously identify different mycoplasma species [24,25,26]. In the Hoechst 33258 stain, the dye selectively binds to DNA, allowing potential contaminants to be visualized under a fluorescence microscope [27]. Luminometric assays are another sensitive method for detection of mycoplasmas [28,29]. In these assays, mycoplasma contamination is detected by adding specific substrates that produce luminescence through the activity of mycoplasma-specific enzymes not present in eukaryotic cells. In this assay, viable mycoplasmas in a test sample (cell supernatant) are lysed, and the released enzymes react with a substrate, converting ADP to ATP. The ATP is then converted into a light signal using the luciferase enzyme. By measuring ATP levels before and after the addition of the substrate in a luminometry system, a ratio can be calculated to indicate the presence or absence of mycoplasma in a sample. Direct visualization through scanning electron microscopy is another option, but this option is elaborate and not suitable for routine usage [29]. This method provides high-resolution, three-dimensional images of cell surfaces, allowing for the visualization of mycoplasma organisms adhering to or invading host cells. It is valuable for identifying mycoplasma contamination by enabling researchers to observe morphological characteristics and spatial relationships within the culture.

By guaranteeing that they send mycoplasma-free cells, these banks help ensure the quality and reliability of the biological materials provided to researchers, thereby supporting the integrity of scientific investigations.

Similarly, viral infection of cell cultures poses significant challenges in research and biotechnology. Contaminating viruses can alter cellular behavior, affect experimental outcomes, and compromise the integrity of results [30,31,32]. These infections can lead to changes in cell morphology, growth rates, and gene expression, ultimately skewing data interpretation. Additionally, viral contamination can impede the production of biopharmaceuticals and vaccines, as it may impact yield and product quality. Therefore, early detection of viral infections is crucial for maintaining healthy cell cultures and ensuring reliable experimental conditions. Most cell banks have implemented stringent monitoring protocols to identify and effectively eliminate potential viral contaminants. In this regard, single and multiplex real-time PCR protocols are essential tools for the detection of viral contaminants in cell cultures [33]. These techniques enable rapid, sensitive, and specific identification of viral nucleic acids, allowing researchers to monitor and control contamination effectively. Single real-time PCR targets a specific virus, providing clear results, while multiplex assays can simultaneously detect multiple viral species in a single reaction, enhancing efficiency and saving time (Figure 5). The quantitative aspect of these assays also allows for the evaluation of viral load, which is essential for assessing the level of contamination.

To prevent sample swapping, cell culture banks have implemented several strategic measures. These include robust labeling systems that incorporate unique identifiers for each sample, rigorous training for staff on the importance of proper handling and documentation, and the use of automated tracking technologies that log every interaction with samples. Additionally, regular audits and quality control checks are conducted to ensure adherence to protocols while maintaining a clear chain of custody for all specimens. By combining these strategies, cell culture banks significantly reduce the risk of sample misidentification and ensure the integrity of their biological resources. In these processes, barcoding is a crucial method for enhancing the efficiency and accuracy of biobanking processes [34]. By assigning unique lab sample barcodes to biological samples, researchers can ensure precise tracking and management throughout storage and analysis. This system minimizes the risk of sample misidentification and contamination, thereby preserving the integrity of the biobank. Additionally, automated barcode scanning systems streamline sample retrieval and data entry, making it easier for scientists to access and utilize the stored resources. Overall, barcoding enhances the traceability and reliability of materials stored in a cell culture bank, facilitating research and improving the quality of scientific data.

A similar barcoding concept was introduced to describe and monitor the cellular diversity of cell clones [35]. This method uses single-copy genomic labeling to track cell populations, which is ideally suited to identifying inter-clonal heterogeneity and sub-lineages of cell clones that exhibit substantial genetic and phenotypic diversity [35]. Since it is known that STR profiling can have insufficient sensitivity in the authentication of biosamples and detection of contamination, the adoption of DNA barcoding technology based on next-generation methods might offer an attractive technology for high-throughput and low-cost authentication, characterization, and contamination testing of cell lines [36]. Due to their proven effectiveness in safeguarding sample integrity, these methods are expected to be seamlessly integrated into the daily workflow of cell banks in the future, enhancing operational efficiency and reliability.

### 2.4. General Documentation

A cell bank repository should offer robust technical support to assist researchers in effectively utilizing the biological materials provided. This support may include guidance on optimal culture practices, troubleshooting common issues related to cell growth and maintenance, and recommendations for specific applications of the cell lines. Additionally, technical staff should be available to address inquiries regarding cryopreservation techniques, thawing protocols, and best practices for handling and storing cells. By providing comprehensive technical assistance, a cell bank can enhance the user experience, facilitate successful experiments, and ultimately contribute to more reliable research outcomes. To facilitate communication, the repository should provide accessible contact information, including email addresses, telephone numbers, and live chat options for real-time assistance. By offering comprehensive technical support and multiple channels for communication, a cell bank can enhance the user experience, facilitate successful experiments, and ultimately contribute to more reliable research outcomes.

In particular, culturing cell lines requires meticulous attention to detail regarding their specific needs—ranging from optimal growth conditions to effective handling practices during passaging and cryopreservation processes [14]. By following well-established protocols and maintaining rigorous quality control measures, researchers can achieve reliable results while ensuring the health of their cultures over time. A cell culture bank must provide comprehensive information regarding the culture conditions of each cell line to ensure optimal growth and functionality. This includes details such as the recommended culture medium, serum requirements, and any specific supplements necessary for maintaining cell viability. Additionally, the bank should specify optimal temperature, humidity, and CO_2_ levels for incubation, as well as passaging techniques and frequency to prevent overgrowth or senescence.

One essential element of this documentation is the Certificates of Analysis (CoA), which confirm the quality and characteristics of each batch of cells supplied. These certificates provide vital information regarding cell viability, sterility, and mycoplasma testing results, giving researchers confidence in the integrity of their biological materials. Additionally, the cell bank should offer references to research publications that have utilized specific cell lines, demonstrating their relevance and applicability in various research contexts. This combination of CoAs and scholarly references not only supports informed decision-making by researchers but also enhances the overall credibility and trustworthiness of the cell bank’s offerings.

### 2.5. Cryopreservation Services

Cryopreservation is a crucial technique in cell culture that allows for the long-term storage of viable cells at ultra-low temperatures. Proper cryopreservation methods are essential to maintain the integrity and functionality of cell lines for future research applications [37]. The ability to store cells in a viable state over extended periods is invaluable in biological research. Cryopreservation not only facilitates the long-term preservation of cell lines but also enables researchers to share resources across laboratories without compromising cell viability. However, improper techniques can lead to cellular damage and loss of functionality. Therefore, information on cryopreservation protocols is also essential for researchers who wish to store cells long-term. By providing this critical data, cell banks enable researchers to replicate optimal conditions in their own laboratories, ensuring consistent and reliable experimental outcomes.

Cell banks commonly store their biological materials, including cell lines and tissues, in controlled environments to ensure long-term viability. One common method involves cryopreservation, where cells are frozen in the presence of a cryoprotective agent at ultra-low temperatures using liquid nitrogen (N_2_) tanks that maintain temperatures around −196 °C, effectively halting metabolic processes and preserving cellular integrity [37]. In these tanks, cells are typically stored in cryovials that contain cryoprotectants to prevent ice crystal formation during freezing (Figure 6).

Additionally, cell banks often use rigorous monitoring systems to track temperature and environmental conditions within the storage facility, ensuring that samples remain safe and viable for future research or therapeutic applications. In this context, passage numbers and lot numbers are crucial for tracking the history and quality of cell lines in research and biobanking. The passage number indicates how many times a cell line has been subcultured, reflecting its growth history and potential genetic drift over time [14]. Higher passage numbers may lead to altered characteristics, which can impact experimental outcomes. Lot numbers, on the other hand, identify a specific batch of cells processed together, ensuring traceability for quality control purposes. Together, these identifiers provide essential information about the cells’ lineage and stability, enabling researchers to choose appropriate samples for their studies.

### 2.6. Ethical Compliance

One crucial element is source transparency, which involves providing clear information regarding the origins of the biological materials used in developing the cell lines. This transparency allows researchers to understand the lineage and characteristics of the cells they are working with. Additionally, ethics approval documentation is vital, serving as evidence that all materials were obtained in accordance with ethical guidelines and regulations. This includes proof of informed consent from donors where applicable and adherence to institutional review board (IRB) protocols. By ensuring both source transparency and robust ethics approval documentation, a cell bank not only upholds scientific integrity but also reinforces public confidence in its practices and offerings.

The principles of the 3Rs—Replacement, Reduction, and Refinement—play a crucial role in establishing immortalized animal cell lines by promoting the use of these alternatives to minimize the reliance on animal models in research [38,39]. By creating and utilizing immortalized cell lines, researchers can reduce the number of animals needed for experiments (Reduction) and refine their methodologies to enhance scientific outcomes while ensuring ethical standards are upheld.

### 2.7. Training Resources

A cell culture repository should ensure that researchers have access to the necessary knowledge and skills for effective cell culture practices. To support this, training resources are essential, including workshops and webinars that provide educational programs covering best practices in cell culture techniques. These interactive sessions allow researchers to engage with experts and gain insights into optimal handling methods, contamination prevention, and troubleshooting strategies. Additionally, online tutorials and guides, such as instructional videos or written manuals, serve as valuable resources for researchers looking to familiarize themselves with specific types of cells or specialized protocols. By offering these comprehensive training resources, a cell bank not only enhances the competency of its users but also promotes successful research outcomes through informed and effective use of their biological materials.

### 2.8. Services Offered for Depositing New Cell Lines in a Cell Bank Repository

When a researcher wishes to deposit a new cell line in a cell bank repository, several key offerings should be provided to facilitate the process. First, the bank should offer clear guidelines and documentation requirements for submission, including necessary information on the cell line’s origin, characterization, and any relevant ethical approvals. Additionally, support in completing the required paperwork and ensuring compliance with regulatory standards is essential. The repository should also provide assistance with testing protocols to verify the quality and viability of the deposited cells before inclusion in their catalog. Finally, an agreement outlining intellectual property rights and usage terms should be established to protect both the researcher’s contributions and the bank’s interests. By offering these comprehensive services, a cell bank can streamline the deposition process while fostering collaboration and innovation in research.

### 2.9. Customization Options for Bespoke Cell Lines

Customization options are an essential aspect of modern cell bank repositories, allowing researchers to tailor biological materials to their specific needs. One key offering is the development of bespoke cell lines, which includes services for creating custom or genetically modified cell lines based on user specifications. These bespoke solutions enable researchers to incorporate specific genetic alterations, reporter genes, or other modifications that align with their experimental objectives. By providing flexibility in cell line creation, cell banks can support a diverse range of research applications and enhance the overall relevance and impact of scientific investigations.

### 2.10. Collaboration Opportunities for Enhanced Research Initiatives

By establishing collaborations with academic institutions or industry players, cell banks can facilitate research initiatives that involve shared resources, expertise, and technologies. These partnerships not only enhance the accessibility of high-quality biological materials but also promote interdisciplinary approaches to tackling complex scientific questions. Through collaborative efforts, researchers can leverage diverse skill sets and knowledge bases, ultimately driving forward the development of new therapies and solutions in various fields of biomedical research.

### 2.11. Shipping and Handling Protocols for Biological Materials

Shipping and handling are critical aspects of cell bank operations, ensuring the safe transport of live cells or frozen samples to researchers worldwide [5]. To maintain the integrity of these biological materials, cell banks employ safe packaging solutions designed specifically for national or international shipping [5,39]. These solutions include insulated containers and cryogenic vials that protect against temperature fluctuations (Figure 7).

When sending cells to national versus international destinations, several key differences must be considered [39]. National shipments typically involve fewer regulatory hurdles and shorter transit times, which can help maintain cell viability and reduce the risk of temperature fluctuations during transport. In contrast, international shipments often require compliance with additional regulations, including customs documentation and import/export permits, which can complicate the process [39]. Moreover, shipping internationally may necessitate specialized packaging to ensure that cells remain viable over longer distances and varying environmental conditions. Understanding these differences is crucial to ensure the successful transfer of biological materials while adhering to legal and safety requirements.

Additionally, temperature-controlled shipping methods are utilized to ensure viability upon arrival, allowing for precise monitoring of conditions throughout transit. Many repositories also offer tracking options, enabling researchers to stay informed about their shipments in real time. Additionally, proper labeling of biological materials is complex due to the stringent regulatory requirements, varying international standards, and necessity for accurate identification of materials throughout the entire logistics process, which must encompass not only the physical protection of samples but also the tracking of environmental factors such as temperature and humidity during transit to maintain sample integrity and compliance with safety protocols. The European Agreement concerning the International Carriage of Dangerous Goods by Road (ADR), the International Air Transport Association Dangerous Goods Regulations (IATA DGR), several international standards developed by the International Organization for Standardization (ISO) such as the ISO 21973 [40], and the EU Tissue and Cells Directive (Directive 2004/23/EC) are typical examples focusing on the safety and quality of biological samples and materials that might present risks to human health or the environment during shipping.

By prioritizing effective shipping and handling practices, cell banks can guarantee that the quality of their products and environments are preserved from the moment potential hazardous samples leave the facility until they reach their final destination.

## 3. Cell Culture Banks

### 3.1. Overview of Notable Cell Bank Repositories Worldwide

There are numerous cell bank repositories worldwide. Notable repositories include the American Type Culture Collection (ATCC) in the United States, which offers an extensive collection of cell lines, microorganisms, and other biological products. The European Collection of Authenticated Cell Cultures (ECACC) focuses on providing well-characterized human and animal cell lines for research across Europe. Additionally, institutions like the Japanese Collection of Research Bioresources (JCRB) serve as vital resources for researchers in Asia. In addition, several lesser-known repositories around the world contribute significantly to the field of biomedical research. For instance, the Korean Cell Line Bank (KCLB) in South Korea specializes in providing a variety of human and animal cell lines, supporting both domestic and international research initiatives. Similarly, the National Institute of Health Sciences (NIHS) in Japan maintains a collection of authenticated cell lines that are pivotal for pharmacological and toxicological studies. In Brazil, the Brazilian Cell Bank (BCR) focuses on preserving native cell lines and promoting research related to local biodiversity. Other notable examples include the Russian Collection of Cell Cultures (RCCC), which offers unique strains for studies in genetics and biotechnology, and the Leibniz Institute DSMZ (German Collection of Microorganisms and Cell Cultures), which is a prominent research institution that serves as a major repository for various biological materials (Figure 8).

Some repositories comply with international standards, such as ISO 9001 [41], ISO 13485 [42], or others. This ensures that these facilities maintain high-quality management systems and adhere to best practices in the handling, storage, and distribution of cell lines. This helps guarantee the integrity, safety, and reliability of these biological materials.

Several prominent companies, such as Sigma-Aldrich, Lonza, PromoCell, and Thermo Fisher Scientific, specialize in selling cell lines and primary cells. These companies typically serve as intermediaries, distributing only the cell lines obtained from certified cell bank repositories. Additionally, they offer a diverse range of biological materials to support research in fields, such as drug development, cancer research, and regenerative medicine. Their products are usually accompanied by detailed documentation to ensure quality and compliance with international standards.

Details about selected cell bank repositories are summarized in Table 1.

### 3.2. Pricing Structures and Terms of Use for Cell Lines from Repositories

It should be noted that significant differences exists in prices and terms of use when ordering a cell line from a specific repository. Prices can vary widely depending on the bank’s location and reputation and the specific type of cell line. Established banks like ATCC may charge a premium for their well-characterized lines, while smaller or regional banks might offer more competitive pricing to attract researchers. In general, additional costs may vary based on whether the cells are primary cultures, immortalized lines, or genetically modified strains.

Importantly, each cell culture bank typically has its own set of terms governing how the cells can be used. Some may allow for commercial applications, while others restrict usage to academic research only. Terms may also specify limitations on redistributing the materials or require that any derived products from the cells be shared back with the bank. In some cases, cell banks request the filling out of a Material Transfer Agreement (MTA) when obtaining a cell line [5,45,46,47]. These MTAs outline the conditions under which biological materials can be transferred between parties. The complexity and stipulations within MTAs can differ significantly among banks. Typically, these MTAs include detailed information about the intended use of the cell lines, ensuring compliance with the bank’s terms and conditions and outlining any restrictions on redistribution or commercialization. Some banks may have straightforward MTAs that facilitate quick access to materials, while others might impose stringent conditions regarding intellectual property rights or publication requirements. Additionally, certain institutions might require acknowledgment in publications resulting from research using their cell lines, whereas others may not.

Variations in quality assurance protocols can also influence pricing and terms; more rigorous testing and characterization processes typically lead to higher costs but provide greater confidence in the reliability of the materials. Finally, shipping costs and import/export regulations can further impact overall expenses and availability of specific cell lines in different regions.

When sending genetically modified cells, it is essential that the bank repository guarantees that all necessary regulatory approvals and documentation are in place, including compliance with biosafety guidelines and any specific restrictions outlined by the cell bank. Additionally, clear labeling and proper packaging must be employed to maintain the integrity of the cells during transit while adhering to applicable shipping regulations for biohazards.

Overall, researchers should carefully review each bank’s offerings and agreements to ensure they align with their project needs and budget constraints before proceeding with a purchase or collaboration.

### 3.3. Balancing Quality and Accessibility in Cell Bank Practices: The Impact of Selling Cells and Redistribution Restrictions

As mentioned earlier, some cell banks that sell biological materials impose restrictions on the redistribution of purchased cells through MTAs, including outright prohibitions on sharing these resources [5,45,46,47]. This practice has both advantages and disadvantages that significantly impact the research landscape. On one hand, by selling cells, these banks can maintain stringent quality assurance standards, effectively controlling the distribution of well-characterized and rigorously tested cell lines. This approach greatly enhances the reproducibility of research outcomes, providing researchers with reliable tools for their investigations. Furthermore, this prohibition on redistribution promotes adherence to ethical standards, ensuring that cells are used only under specified conditions and with the necessary approvals. Financial sustainability is another advantage; by generating revenue through sales, cell banks can reinvest in maintaining their collections, conducting research, and developing new cell lines. Additionally, controlled distribution ensures clear documentation regarding the origin and usage of cells, which is crucial for projects requiring transparency. On the other hand, there are notable drawbacks to this model. The costs associated with purchasing cells may pose a barrier for smaller laboratories or research institutions, limiting access to essential biological materials. Moreover, restrictions on sharing or redistributing cell lines could hinder innovative approaches by reducing collaboration between different institutions. There are initiatives for relaxing restrictions on the redistribution and commercial use of biomaterials while maintaining aspects of standard MTAs [47]. Researchers may also become overly reliant on specific cell banks; this dependence can be problematic if those banks alter their services or cease operations altogether. Finally, prohibitions against redistribution may introduce extra legal and administrative requirements that could slow down the research process.

In conclusion, while selling cells with restrictions on redistribution helps ensure quality and ethical compliance in research practices, it is essential for cell banks to strike a balance between these factors and maintaining accessibility to foster innovation in scientific inquiry.

### 3.4. Enhancing Cell Bank Services: Opportunities for Improvement and Addressing Limitations

Cell banks have significant potential to enhance their services and impact in several key areas. First, improving accessibility is crucial. By offering tiered pricing structures or discounts for smaller laboratories and academic institutions, cell banks can make high-quality biological materials more accessible to a broader range of researchers [1]. Additionally, expanding educational resources through comprehensive training programs, workshops, and online tutorials on best practices in cell culture techniques would empower researchers to utilize cell lines effectively and reduce contamination risks [48,49,50].

Moreover, enhancing customization options by providing bespoke services for creating custom or genetically modified cell lines based on specific research needs would allow for greater flexibility and innovation in scientific studies. Streamlining the MTA process and making it more transparent could facilitate quicker access to cells while ensuring compliance with ethical standards [47]. Furthermore, implementing even more rigorous quality control measures—such as regular testing for contaminants like mycoplasma—would further enhance the reliability of the provided cell lines.

Collaboration opportunities also play a vital role. Actively fostering partnerships with academic institutions and industry players can lead to shared resources, knowledge exchange, and collaborative research initiatives that benefit all parties involved. Developing user-friendly online platforms for ordering cells, tracking shipments, and accessing documentation could significantly improve the overall user experience.

However, there are additional limitations that cell banks must address. For instance, many researchers face challenges related to the limited availability of certain rare or specialized cell lines. Additionally, the complexity of regulatory compliance can be daunting for some users unfamiliar with the necessary protocols. This complexity arises from the intricate web of regulations and standards that govern research practices, particularly in highly regulated fields such as biotechnology, pharmaceuticals, and medical research. Users may face a myriad of guidelines set forth by various regulatory bodies, including local, national, and international entities, each with its own specific requirements concerning safety, quality control, documentation, and reporting. Undoubtedly, for those who are not well-versed in these regulations, the task of navigating this landscape can be overwhelming. This highlights the necessity for robust training programs and resources to help researchers and industry professionals understand and adhere to compliance protocols. The support of knowledgeable personnel, along with accessible guidelines and tools, can empower users to confidently engage with regulatory requirements. Communication barriers may arise if customer support is not readily available or responsive to inquiries about specific products or services.

## 4. Conclusions

Cell banks are facilities that perform a variety of activities typical of standard laboratories, but on a larger scale (Appendix A). They are crucial for advancing biomedical research, providing standardized and well-characterized biological materials essential for a range of applications. While the practice of selling cells and imposing restrictions on their redistribution through MTAs offers advantages, such as improved quality assurance and ethical compliance, it also presents challenges related to accessibility and collaboration. To better support the diverse needs of researchers and foster innovation, cell banks should focus on improvements in several key areas, including pricing structures, educational resources, customization options, and user-friendly platforms. By addressing existing limitations, such as the availability of specialized cell lines and streamlining regulatory processes, cell banks can enhance their impact on scientific discovery. In summary, we hope our article will provide valuable insights for the scientific community by synthesizing knowledge about cell banks and their essential role in biomedical research. It highlights the dual nature of cell sales, balancing quality assurance with challenges in accessibility, and identifies critical areas for improvement. Ultimately, by promoting collaboration between cell banks and researchers, we can drive advancements in healthcare and therapeutic development. Through these efforts, cell culture banks will continue to be indispensable resources, supporting researchers in their pursuit of knowledge and breakthroughs in biomedicine.

## Figures and Tables

**Figure 1 cells-13-01861-f001:**
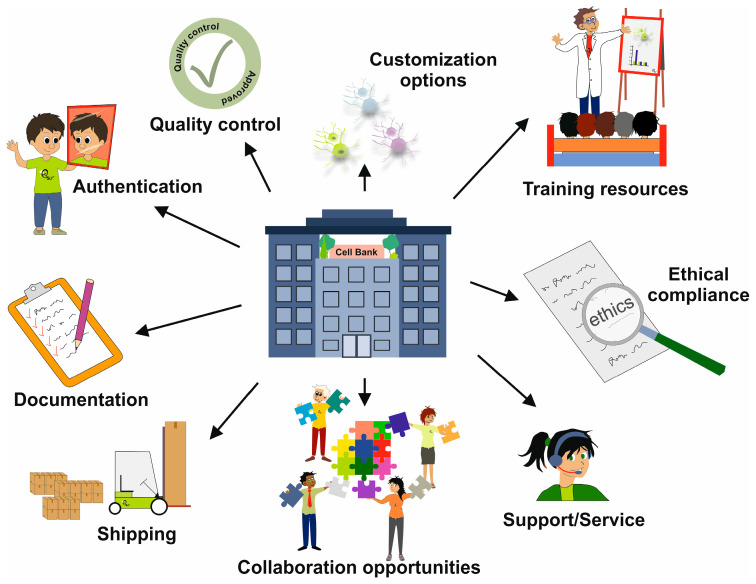
Overview of the services that a cell bank should provide. A cell bank should offer various cell types from different species, such as primary cells, immortalized cell lines, and stem cells, tailored for specific applications. In addition, a cell bank should offer consultation, relevant information, and a range of support services to researchers.

**Figure 2 cells-13-01861-f002:**
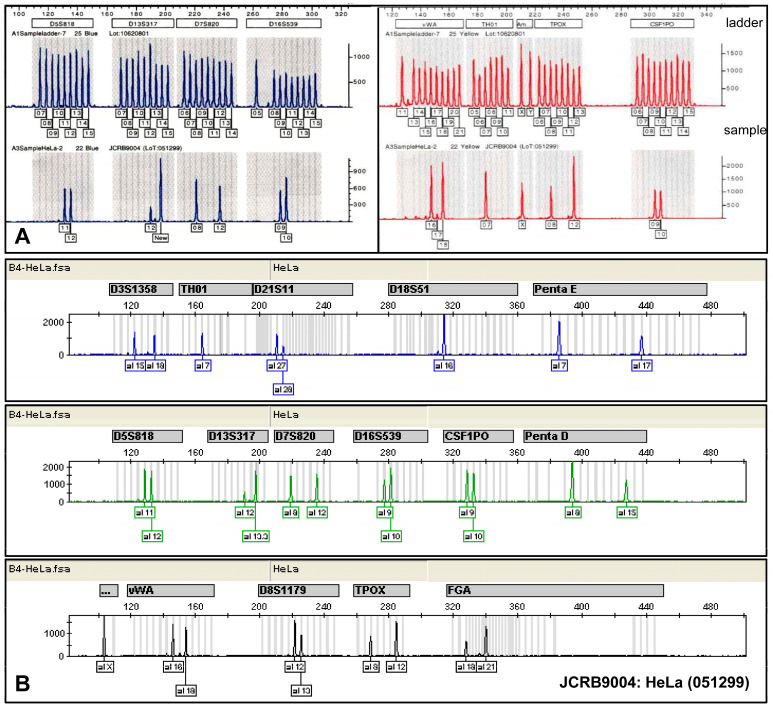
STR profiling of HeLa cells. Two different panels of STR markers were used for the short tandem repeat profiling of HeLa cells from the Japanese Collection of Research Bioresources (JCRB). In panel (**A**), the STR panel is based on nine variant markers (D5S818, D13S317, D7S820, D16A539, vWA, TH01, Amelogenin, TPOX, and CSF1PO) which were amplified, and the sizes of the resulting amplicons were compared to ladders containing all known variant sizes. Panel (**B**) includes the eight markers D5S818, D13S317, D16A539, vWA, TH01, Amelogenin, TPOX, and CSF1PO from panel (**A**) along with seven additional markers (D3S1358, D21S11, D18S51, Penta E, Penta D, D8S1179, and FGA). This comparative analysis demonstrates the distinct allelic patterns generated by each profiling system, confirming the genetic identity of HeLa cells. The allelic patterns for HeLa cells are as follows: D5S818 (11,12), D13S317 (12,13.3), D7S820 (8,12), D16S539 (9,10), vWA (16,18), TH01 (7), Amelogenin (X), TPOX (8,12), CSF1PO (9,10), D3S1358 (15,18), D21S11 (27,28), D18S51 (16), Penta E (7,17), Penta D (8,15), D8S1179 (12,13), and FGA (18,21), respectively.

**Figure 3 cells-13-01861-f003:**
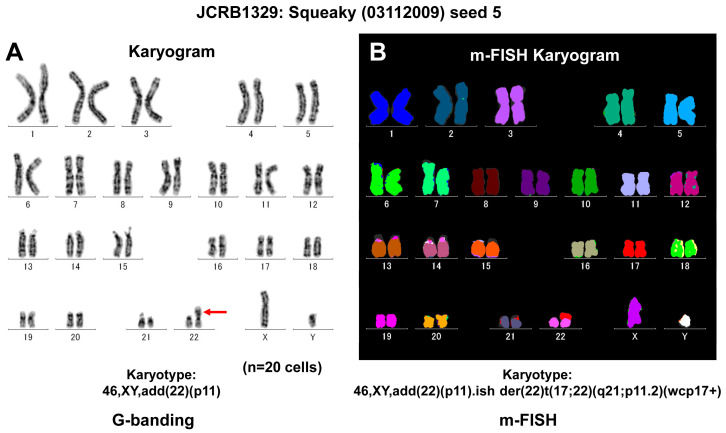
Genetic characterization of the human transformed cell line known as Squeaky. Squeaky is a cell line similar to embryonic stem cells, obtained from the lung of a 14-week-old male fetus (MRC-5). This cell line was infected with a recombinant retrovirus that expressed the four factors Oct3/4, Sox2, Klf4, and c-Myc, resulting in an infinite lifespan. (**A**) G-banding analysis confirmed a male karyotype with a normal number of chromosomes, but an additional piece of genetic material was identified on chromosome 22 at the p11 region (marked with red arrow). (**B**) Further analysis using Multiplex Fluorescence In Situ Hybridization (m-FISH) revealed a derivative chromosome 22, which was a result of a translocation with chromosome 17, specifically affecting regions q21 and p11.2. This translocation can be best visualized using a whole chromosome probe for chromosome 17 (wcp17+).

**Figure 4 cells-13-01861-f004:**
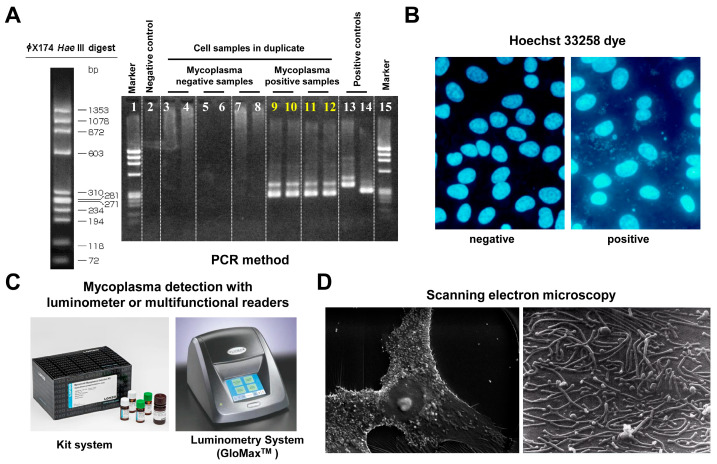
Mycoplasma testing. There are several methods available for detecting mycoplasma contamination. (**A**) Mycoplasma detection by PCR involves amplifying specific DNA sequences unique to mycoplasma species, allowing for sensitive and rapid identification of contamination. In the depicted experiment, universal primers specific for most mycoplasma species were used. (**B**) Mycoplasma detection using the Hoechst 33258 bisbenzimide dye stain involves fluorescent staining of DNA, allowing for the visualization of mycoplasma cells under a fluorescence microscope. This method is advantageous because it can differentiate between eukaryotic and prokaryotic cells based on their size and morphology, providing a quick assessment of contamination in cell cultures. (**C**) The MycoAlert™ Mycoplasma Detection Kit is based on a selective biochemical test that exploits the activity of mycoplasma enzymes found in all six major mycoplasma cell culture contaminants and the vast majority of the 180 mycoplasma species. (**D**) Mycoplasma detection by scanning electron microscopy (SEM) allows for high-resolution imaging of mycoplasma cells, enabling observation of their unique morphology and attachment to host cells.

**Figure 5 cells-13-01861-f005:**
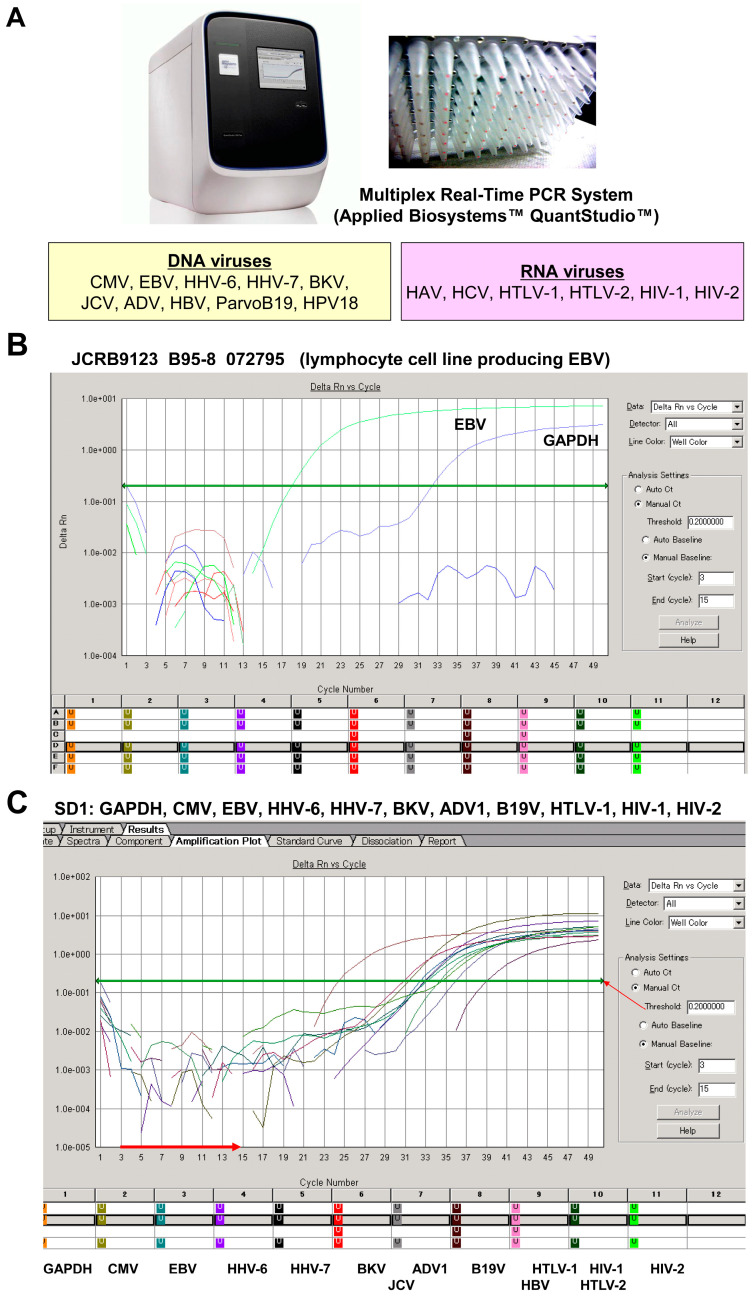
Multiplex PCR analysis of viral detection. (**A**) The setup of the multiplex real-time PCR system uses specific primers and probes to simultaneously amplify target sequences specific to DNA viruses (CMV, EBV, HHV-6, HHV-7, BKV, JCV, ADV, HBV, ParvoB19, and HPV18) as well as RNA viruses (HAV, HCV, HTLV-1, HTLV-2, HIV-1, and HIV-2). (**B**) The Epstein–Barr virus (EBV)-transformed lymphocyte cell line B95-8, which originates from *Saguinus oedipus* (cotton-top tamarin), was tested for the presence of EBV. The analysis confirms distinct amplification patterns indicative of EBV presence, with the housekeeping control GAPDH used as a reference; no other viruses were detected. (**C**) A standard sample (SD1) containing nucleic acids from various viruses underwent multiplex real-time PCR. This assay shows that the experimental setup can detect multiple viruses simultaneously. Further details of this assay can be found elsewhere [33]. The horizontal red arrow indicates the zero line, while the green horizontal arrow indicates the cut-off line. Panels (**A**,**B**) are screenshots from the multiplex real-time PCR system.

**Figure 6 cells-13-01861-f006:**
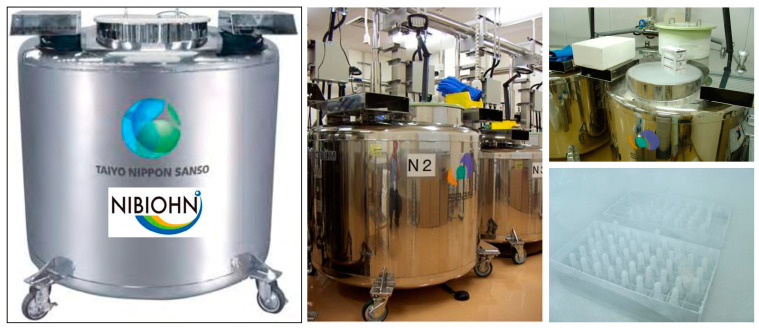
Cell storage. The image depicts multiple liquid nitrogen (N_2_) tanks, each containing cryovials with frozen cells preserved at ultra-low temperatures. The frozen cells are stored in boxes within the vapor phase of the nitrogen, ensuring optimal conditions for maintaining viability and integrity over extended periods.

**Figure 7 cells-13-01861-f007:**
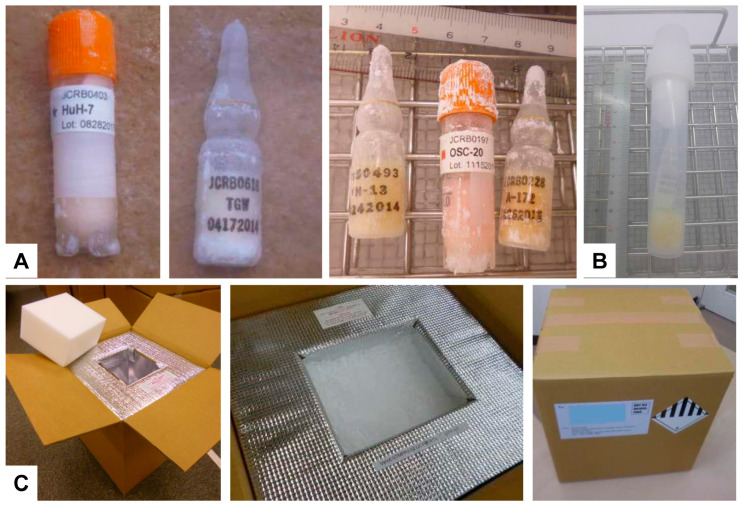
Parceling of frozen cells for transport. (**A**) Various glass or plastic storage vials containing frozen cells prepared for dispatch are depicted. (**B**) A secure container designed to hold the vials during transit is shown. (**C**) The packaging process is highlighted, emphasizing the use of dry ice to maintain the low temperatures essential for preserving cell viability.

**Figure 8 cells-13-01861-f008:**
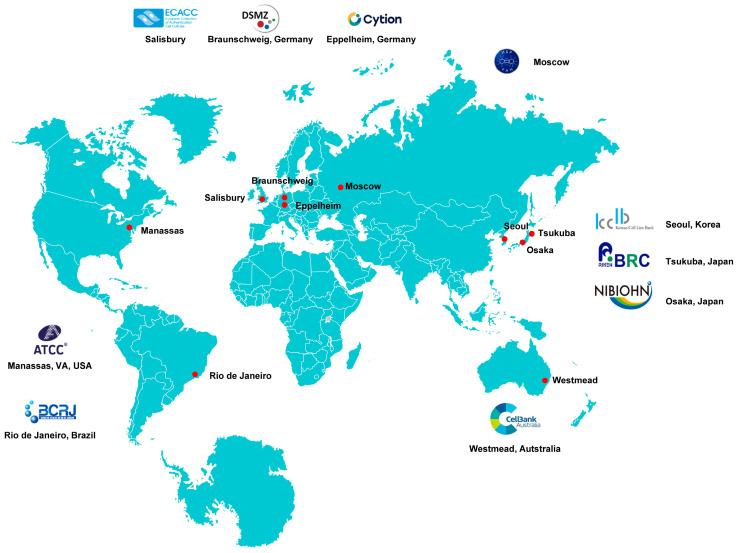
Locations of representative cell culture biobanks worldwide. Depicted are various sites across America, Europe, Asia, and Australia where cell culture biobanks have been established. Each marked location represents a significant hub for cellular research and biobanking activities, contributing to advancements in biomedical science and regenerative medicine. The map highlights the global distribution of these facilities, emphasizing their importance in fostering international collaboration and innovation in the field of cell biology.

**Table 1 cells-13-01861-t001:** Summary of key information on selected cell bank repositories.

Cell Bank Name	Location	Type of Cell Lines Offered	MTA	Quality Assurance Standards	Collaboration Opportunities	Contact Information ^1^
ATCC	USA	>4000 human and animal cell lines	required	ISO 9001 [41] ^2^ ISO 13485 [42] ^3^, ISO/IEC 17025 [43] ^4^ ISO 17034 [44] ^5^	Partnership with several companies and institutes (InSphero, USP, Synthgo, Qiagen, One Codex, NIST, and LGC Standards)	10801 University Boulevard, Manassas, Virginia 20110-2209, USA Phone: (703) 365-2700 http://www.atcc.org ^8^ Email via contact form
ECACC	UK	>1100 cell lines from over 45 species	required	ISO 9001 [41] ISO/IEC 17025 [43]	Partnerships with many local distributors (Merck, KAC, and Cell Bank Australia)	UK Health Security Agency, Porton Down, Salisbury, SP4 0JG, UK Phone: +44 (0)1980 612512 https://www.culturecollections.org.uk/ ^8^ Email: culturecollections@ukhsa.gov.uk
JCRB ^6^	Japan	1642 cell lines, of which 1111 are of human origin	request and agreement form required	Testing for bacteria, fungi, mycoplasma, and viruses; performing species identification, cell identification, and chromosome analysis	Teamed up with the National Institute of Biomedical Innovation, Health and Nutrition (NIBIOHN)	7-6-8 Saito-Asagi, Ibaraki Osaka 567-0085, Japan https://cellbank.nibiohn.go.jp/english ^8^ Email: jcrb-cell@nibiohn.go.jp
DSMZ	Germany	>900 from various species	required	Testing for growth characteristics, bacteria (notably mycoplasms), fungi, yeast, and human pathogenic viruses; species identification and authentication	Member of several national and international organizations, networks, and projects	Inhoffenstraße 7B 38124 Braunschweig Science Campus Braunschweig-Süd, Germany Phone: +49 (0)531 2616-0 https://www.dsmz.de/ ^8^ Email via contact form
KCLB	South Korea	423 cell lines from various human tissues (gastric, colon, lung, cervical, ovarian, pancreas, breast, and other cancer cell lines)	required	Scientific quality control including STR profiles	Contract research organization service, hands-on workshops	Cancer Research Institute, Seoul National University College of Medicine, 103, Daehak-ro, Jongno-gu, Seoul, Republic of Korea, 03080 Phone: +82-02-3668-7915 Email: kclb@kclb.kr https://cellbank.snu.ac.kr/eng/ ^8^
BCRJ ^7^	Brazil	Primary cells and immortalized cell lines	NN	Several services including cell storage, screening for mycoplasma, toxicity tests, cell immortalization, and cell authentication	Offers courses in basic and good practices in cell culture	Av. N. S. das Gracas, 50, Prédio 32, Parque Tecnológica de Xerém Duque de Caxias, Rio de Janeiro, Brazil Phone: +55 21 2145-3337 https://bcrj.org.br/ ^8^ bcrj@bcrj.org.br
Cytion	Germany	>800 human and animal cells, stem cells, and primary cells	not required	ISO 9001 [41] Mycoplasma testing via colorimetric assay and a PCR-based method, STR analysis, and testing for viral/bacteria/fungi contaminants	Collaborations with industry and academic institutions	CLS Cell Lines Service GmbH Dr.-Eckener-Str. 8 69214 Eppelheim, Germany https://www.cytion.com/ ^8^ Phone: +49 (0)6221 405780 Email: info@cytion.com
RCCC	Russia	~150 mammalian cell lines (mouse, dog, rabbit, rat, monkey, pig, and hamster; human primary and cancer cells	NN	Cells are tested for contamination; STR profile	Associated with the Koltzov Institute of Development Biology which offers Doctoral/PhD programs	Koltzov Institute of Developmental Biology of the Russian Academy of Sciences 26 Vavilov Street, Moscow Phone: +7 (499) 135-33-22 Email via contact form
RIKEN BRC	Japan	NN	required, some cells with restrictions	ISO 9001 [41] Provides validation reports on request; started with cell verification testing service	Provides annual technical training course for researchers, students, and technicians; offers cooperation (e.g., deposition of cells)	Riken BioResource Center 3-1-1 Koyadai, Tsukuba, Ibaraki, 305-0074, Japan https://cell.brc.riken.jp/en/ ^8^ Email: cellqa.brc@riken.jp
CellBank Australia	Australia	>1300; distributes cell lines from ECACC and JCRB. In addition, several mouse and human cancer cell lines are distributed	required	ISO 9001 [41] ISO/IEC 17025 [43]	Partnership with ECACC and JCRB	Children’s Medical Research Institute 214 Hawkesbury Road Westmead NSW 2145 Locked Bag 23 Phone: +612 8865 2850 https://www.cellbankaustralia.com/ ^8^ Email: cellbank@cmri.org.au

^1^ Many of the websites of the listed companies provide a wealth of informative resources, offering valuable insights into their products, services, and advancements in the field. ^2^ ISO 9001 is a globally recognized standard for Quality Management Systems (QMS) that demonstrates the ability to consistently provide products and services that meet customer and regulatory requirements. ^3^ ISO 13485 is an internationally agreed standard for QMS in the medical device industry, which ensures that all medical devices meet the proper regulatory compliance laws and customer needs. ^4^ ISO/IEC 17025 is the worldwide international standard for testing and calibration laboratories, indicating that requirements for the competence, impartiality, and consistent operation of laboratories is fulfilled and ensuring the accuracy and reliability of their testing and calibration results. ^5^ ISO 17034 is the basic standard for reference material producers that define the requirements that must be met for the competent production of high-quality and certified reference materials. ^6,7^ Some impressions from the JCRB and the BCRJ are given in Appendix A. ^8^ last accessed 9 November 2024. Abbreviations used: ATCC, American Type Culture Collection; BCRJ, Banco de Células do Rio de Janeiro; DSMZ, Deutsche Sammlung von Mikroorganismen und Zellkulturen; ECACC, European Collection of Authenticated Cell Cultures; ISO, International Organization for Standardization; JCRB, Japanese Collection of Research Bioresources, KCLB, Korean Cell Line Bank; MTA, Material Transfer Agreement; NIHS, National Institute of Health Sciences; NIST, National Institute of Standards and Technology; NN, not known; RCCC, Russian Collection of Cell Cultures; USP, US Pharmacopeia.

## Data Availability

No new data were created or analyzed in this study. Data sharing is not applicable to this article.

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
