# Peer review of "Unlocking Potential: A Comprehensive Overview of Cell Culture Banks and Their Impact on Biomedical Research"

_cells, 2024, doi:10.3390/cells13221861_

Round 1

Reviewer 1 Report

Comments and Suggestions for Authors

Review in attachment

Author Response

Dear Reviewer 1,

Thank you very much for taking the time to read our manuscript and for your encouraging remarks. Please find our response to your suggestions in the attached pdf-file.

Regards

Ralf Weiskirchen

Reviewer 2 Report

Comments and Suggestions for Authors

In this work, the authors provided a comprehensive overview of Cell Culture Banks (CCB) and their impact on biomedical research. The article is the authors' opinion in which they highlight the differences between the various CCB and discuss the technological advances and methods used by these banks to ensure and maintain the high quality of stored cell culture banks. In addition, the authors discussed the challenges researchers face in accessing high-quality cell lines and proposed strategies to improve collaboration between academic institutions and commercial players. This work is interesting and informative. Publication is warranted if the following concerns are addressed:

Some minor issues:

1)        Page 14, Table 1, first line – there is an abbreviation ‘MTA’, which is explained in the text later, as late as page 16.

2)        Page 14 (and 15), Table 1, last line (and second and third lines from the end) – there is an abbreviation ‘NN’ which is not explained in the text and/or description in the table

3)        Page 15, Table 1, last line, second column – there is only information >1300 - Is there no information available on the type of cell lines offered by CellBank Australia to add here?

4)        In line 415 and 515 and there is a reference to table S1 and S2 - the tables so labelled do not appear in the paper.

It is worth noting further that of the 39 articles cited in the references section, 15 items are from the last five years and no excessive self-citation was found.

Author Response

Dear Reviewer 2,

Thank you very much for taking the time to read our manuscript and for your encouraging remarks. Please find our response to your suggestions in the attached pdf-file.

Regards

Ralf Weiskirchen

Reviewer 3 Report

Comments and Suggestions for Authors

Here are some comments that could be used to recommend acceptance for the manuscript titled "Unlocking Potential: A Comprehensive Overview of Cell Culture Banks and Their Impact on Biomedical Research":

This work comprehensively covers cell culture banks' relevance and various roles in biomedical research. The extensive coverage of quality control, ethics, and biobanking technology benefits both novice and expert readers.

The authors organized the article logically, making complicated themes easier to understand. Each segment builds on the previous, creating a compelling story.

The growing importance of repeatability in biomedical research makes this work contemporary and important. It covers cell banking essentials for high-quality, consistent biomedical research.

Current advances like automated cell banking and customizable cell lines offer useful perspectives on future paths, making the article a great resource for cellular biology research.

The paper establishes a foundation for academic-industry collaboration that could lead to therapeutic development breakthroughs.

Author Response

Dear Reviewer 3,

Thank you very much for taking the time to read our manuscript and for your encouraging remarks. Please find our response to your suggestions in the attached pdf-file.

Regards

Ralf Weiskirchen

Round 2

Reviewer 1 Report

Comments and Suggestions for Authors

Dear authors,

Thank you very much for taking into account the changes in the text and reasonably explaining the doubts.